

# Head-on collision of internal waves with trapped cores

Vladimir Maderich[1], Kyung Tae Jung[2], Kateryna Terletska[1], and Kyeong Ok Kim[2]

[1]Institute of Mathematical Machine and System Problems, Glushkov av., 42, Kiev 03187, Ukraine
[2]Korea Institute of Ocean Science and Technology, 787, Haean-ro, Ansan 426-744 Republic of Korea

*Correspondence to:* Maderich V. (vladmad@gmail.com)

**Abstract.** The dynamics and energetics of a head-on collision of internal solitary waves (ISWs) with trapped cores propagating in thin pycnocline were studied numerically within the framework of the Navier-Stokes equations for a stratified fluid. The peculiarity of this collision is that it involves the trapped masses of a fluid. The interaction of ISWs differs for three classes of ISWs: (i) weakly nonlinear waves without trapped cores, (ii) stable strongly nonlinear waves with trapped cores, and (iii) shear unstable strongly nonlinear waves. The wave phase shift grows as the amplitudes of the interacting waves increase for colliding waves of classes (i) and (ii) and remains almost constant for those of class (iii). The excess of the maximum runup amplitude over the sum of the amplitudes of colliding waves almost linearly increases with increasing amplitude of the interacting waves belonging to classes (i) and (ii); however, it decreases somewhat for those of class (iii). The waves of class (ii) with a normalized on thickness of pycnocline amplitude lose fluid trapped by the wave cores in the range approximately between 1 and 1.75. The interacting stable waves of higher amplitude capture cores and carry trapped fluid in opposite directions with little mass loss. The collision of locally shear unstable waves of class (iii) is accompanied by the development of three-dimensional instability and turbulence. The dependence of loss of energy on the wave amplitude is not monotonous. Initially, the energy loss due to the interaction increases as the wave amplitude increases. Then, the energy losses reach a maximum due to the loss of potential energy of the cores upon collision and then start to decrease. With further amplitude growth, collision is accompanied by the development of instability and an increase in the loss of energy. The collision process is modified for waves of different amplitudes because of the exchange of trapped fluid between colliding waves due to the conservation of momentum.

## 1 Introduction

Internal solitary waves (ISWs) are widespread in stratified oceans and lakes (Helfrich and Melville, 2006). The observed ISWs are mostly waves of mode-1 propagated as waves of depression or as waves of elevation. When near-surface or near-bottom layers are stratified, then mode-1 ISWs of large amplitude can trap and transport fluid in their cores, as observed in the ocean (Moum et al., 1990; Lien et al., 2012; Klymak et al., 2003; Scotti and Pineda, 2004) and in the atmospheric boundary layer (the phenomenon known as "the morning glory" (Christie, 1992; Reeder et al., 1995)). These waves have been studied in laboratory experiments (Grue et al., 2000; Carr et al., 2008; Luzzatto-Fegiz and Helfrich, 2014) . The fluid can also be trapped by ISWs of mode-2 (Yang et al., 2010; Shroyer et al., 2010; Ramp et al., 2015) propagating in the thin interface layer between two uniform density layers, as has been shown in laboratory experiments (e.g., Davis and Acrivos (1967); Maxworthy (1980); Kao and Pao (1980); Honji et al. (1995); Stamp and Jacka (1995); Maderich et al. (2001); Brandt and Shipley (2014); Carr et al. (2015)).



The weakly nonlinear solution for the corresponding ISW (i.e., the Benjamin-Ono (BO) soliton (Benjamin, 1967; Ono, 1974)) agrees well with experimental data for a small amplitude ISW without mass transport. However, experiments and solutions both of the Dubreil-Jacotin-Long (DJL) equation and within the framework of the Navier-Stokes equations (Lamb, 2002; Helfrich and White, 2010; Lamb and Farmer, 2011; Salloum et al., 2012; Maderich et al., 2015; Luzzatto-Fegiz and Helfrich, 2014; Deepwell and Stastna, 2016) show that BO solitons cannot even qualitatively describe the dynamics and transport features of large amplitude waves. A detailed review of laboratory and numerical experiments is given by (Maderich et al., 2015). Little is known regarding the interaction of waves with trapped cores. This kind of interaction is of special interest, as the masses of fluid trapped by waves are also involved in the interaction. The oblique interaction of "morning glories" over northern Australia was described by Reeder et al. (1995). A mostly qualitative description of the head-on collision of mode-2 waves with trapped cores, obtained through conducting several laboratory experiments (Kao and Pao, 1980; Honji et al., 1995; Stamp and Jacka, 1995) and via a numerical simulation (Terez and Knio, 1998), has been given. These experiments showed that (i) waves experience a phase shift during collision, (ii) large amplitude waves in the interaction process exchange trapped masses of fluids between incident waves, and (iii) some trapped fluid is ejected upon collision. According to Gear and Grimshaw (1984), interaction processes can be distinguished as strong interactions when waves propagate in almost the same direction and the time of interaction is relatively long and as weak interaction when waves propagate in almost opposite directions and the time of interaction is relatively short. However, a numerical study of both overtaking and head-on collisions of large amplitude mode-1 and weakly nonlinear mode-2 ISWs Stastna et al. (2015) showed that these interactions are strong interactions resulting in the degeneration of mode-2 ISWs. In this paper, the dynamics and energetics of a head-on collision of ISWs with trapped cores for a wide range of amplitudes and stratifications are studied numerically within the framework of the Navier-Stokes equations. The paper is organized as follows: the numerical flume setup is described in section 2, the results of experiments on the collision of waves of equal and non-equal amplitudes are discussed in section 3, and the main results are summarized in section 4.

## 2   The numerical model setup

A free-surface non-hydrostatic numerical model for variable-density flows using the Navier-Stokes equations in the Boussinesq approximation (Kanarska and Maderich, 2003; Maderich et al., 2012) was applied in the simulations of a numerical flume emulating a laboratory flume filled with salinity-stratified water. The numerical flume and experimental configurations are shown in Fig. 1. Here, $(x, y, z)$ are the Cartesian coordinates in the longitudinal, transverse and vertical directions, respectively. The vertical coordinate $z$ is directed upward. The flume has a length $L_x = 3$ m, a depth $H$=0.46 m and a width $L_y = 0.5$ m. It was filled with water, in which the density of the upper layer is $\rho_1$, and with a thin pycnocline near the bottom, expressed a

$$\rho(z) = \rho_0 \left( 1 - \frac{\Delta \rho}{\rho_0} \tanh \left( \frac{z}{h} \right) \right), \tag{1}$$

where $\rho_0$ is density at the bottom, $\Delta \rho = \rho_0 - \rho_1$ is the density difference between the bottom and the surface, and $h$ is the pycnocline layer thickness. The ISWs were generated at both ends of the flume by the collapse of the mixed regions with the



density $\rho_0$. Following Maderich et al. (2015), the shape of the mixed region was selected to be half of a BO soliton to reduce the mixing due to the collapse. The wave amplitude varied according to the initial volume of the mixed fluid and thickness of the pycnocline $h$. The kinematic viscosity $\nu$ was $1.14 \times 10^{-6}$ m$^2$s$^{-1}$, and the molecular diffusion $\chi$ was $10^{-9}$ m$^2$s$^{-1}$.

An ISW is characterized by an amplitude $a$, which represents the maximum displacement of the isopycnal (Fig. 1), a wave speed $U_c$, calculated as the velocity of the wave crest, and a wavelength $\lambda_{0.5}$, estimated to be a half-width with which the amplitude of the wave is reduced by half. The maximal speed of the wave is $U_m$. The wave parameters were evaluated in the sections $x_L$=$x_R$=0.5 m away from the centre of the laboratory tank $x_C$. For example, the amplitudes of waves propagating from the left to the right in the cross-sections $x_L$ and $x_R$ are defined as $a_L(x_L)$ and $a_L(x_R)$, while those propagating from the right to the left are defined as $a_R(x_L)$ and $a_R(x_R)$, respectively. The simulation results are presented in dimensionless

form. The coordinates are normalized to $h$, and the time $\tau = t/\tau_0$ is dimensionless, where $t$ is time, $\tau_0 = \sqrt{2\rho_0 h/\Delta\rho g}$, and $g$ is gravity; the velocity $\boldsymbol{U} = (U, V, W)$ is normalized to the long linear wave phase velocity $c_0 = \sqrt{gh\Delta\rho/2\rho_0}$ (Benjamin, 1967). The important dimensionless parameters characterizing the waves are the ratio $\varepsilon = H/h$ and dimensionless amplitude $\alpha = a/h$. The Froude number $\mathrm{Fr}_{\max}$ is defined as the ratio of the maximum local velocity $U_m$ to the phase velocity $U_c$:

$$\mathrm{Fr}_{\max} = \frac{U_m}{U_c}. \tag{2}$$

The shear stability of an ISW can be described by the minimum Richardson number $\mathrm{Ri}_{\min}$ calculated for a wave crest:

$$\mathrm{Ri}_{\min} = \frac{g}{\rho_0} \frac{\partial \rho}{\partial z} \left/ \left(\frac{\partial U}{\partial z}\right)^2 \right., \tag{3}$$

where $\rho(x, y, z, t)$ is the density. The wave Reynolds number is defined as

$$\mathrm{Re_m} = \frac{U_m h}{\nu}. \tag{4}$$

The interacting ISWs are given in Table 1. They can be categorized according to the values of the parameters $\mathrm{Fr}_{\max}$ and

$\mathrm{Ri}_{\min}$ (Maderich et al., 2015) into three classes: (i) the weakly nonlinear waves without trapped cores at $1 < \mathrm{Ri}_{\min}$, $\mathrm{Fr}_{\max} < 1$; (ii) the stable strongly nonlinear waves with trapped cores at $0.15 < \mathrm{Ri}_{\min} < 1$, $1 < \mathrm{Fr}_{\max} < 1.3$ ; (iii) the unstable strongly nonlinear waves at $\mathrm{Ri}_{\min} < 0.1$; $\mathrm{Fr}_{\max} \approx 1.35$. The boundary conditions on the surface include the kinematic condition for the free surface. At the lateral and bottom boundaries, the free-slip conditions are used. For large $\varepsilon$, this allows for the simulation of the interaction of mode-1 ISWs with trapped core, propagating in stratified layers near the surface and near the

bottom, and the interaction of mode-2 ISWs, assuming symmetry in the Boussinesq approximation around the horizontal midplane (Maderich et al., 2015). No flux condition for salinity was applied at all boundaries. The model is described in detail in (Kanarska and Maderich, 2003; Maderich et al., 2012) . A total of 35 runs were performed in Series A-D. Most of the runs were performed in a two-dimensional setting with a grid resolution of $3000 \times 400$ (length and height, respectively), whereas several three-dimensional runs were carried out with a grid resolution of $2000 \times 200 \times 45$ (length, height and width, respectively).



## 3 Results and discussion

### 3.1 Interaction of waves of equal amplitudes without trapped cores

The interaction of ISWs of equal amplitude $\alpha_L = \alpha_R = 0.81$ (case (A2; A2)) is shown in Fig. 2a. These waves belong to the class of weakly nonlinear waves without trapped cores ($\mathrm{Fr_{max}}$=0.71, $\mathrm{Ri_{min}}$ = 52 ). After a collision, waves retain their profile and lose amplitude mainly due to the viscous effects. Before and after collision, the wave profiles are similar to those of the weakly nonlinear BO solitons even if the wave amplitudes are not small (Fig. 3). The collision for this class of waves is not fully elastic, as seen in Fig. 4, where the relative excess of the maximum runup amplitude $\alpha_m$ over the sum of the amplitudes of equal interacting waves $\Delta\alpha$ (runup excess) and the phase shift $\Delta\theta$ are plotted versus $\alpha$. Here, $\alpha = \alpha_L = \alpha_R$, $\Delta\alpha = \alpha_m/\alpha - 2$ and $\Delta\theta$ is the phase shift normalized at the characteristic time scale $\tau_0$. The presence of a phase shift due to the collision of mode-2 waves for $\alpha = 0.98$ was confirmed in a laboratory experiment (Honji et al., 1995). As seen in Fig. 4 the relative runup excess $\Delta\alpha$ and phase shift $\Delta\theta$ increase as the interacting wave amplitude $\alpha$ increases.

### 3.2 Interaction of waves with a trapped core and moderate amplitude

The head-on collision between ISWs of equal moderate amplitude with trapped cores $\alpha = \alpha_L = \alpha_R = 1.6$ (case (A5; A5)) is characterized by special features, as seen in Fig. 2b. These waves, belonging to class (ii), i.e., stable strongly nonlinear waves with trapped cores ($\mathrm{Fr_{max}}$=1.11, $\mathrm{Ri_{min}}$=1.1), carried fluid in the cores before collision. However, after collision, the waves lost fluid trapped by the wave cores. This fluid slowly collapsed in the viscous and diffusive-viscous regimes (Galaktionov et al., 2001). The profile of the incident wave at $\alpha = 1.6$ (as well as other characteristics (Maderich et al., 2015)) essentially differs from the predictions made by using the weakly nonlinear theory (Fig. 5a). The amplitudes of transmitted waves ($\mathrm{Fr_{max}}$=1.0, $\mathrm{Ri_{min}}$ =1.2) decrease after collision. They propagate as weakly nonlinear BO solitons (Fig. 5b). This kind of head-on collision occurs in the range of approximately $1 \leq \alpha \leq 1.6$. Notice that in the numerical study Terez and Knio (1998), the wave lost trapped fluid in the process of interaction even at $\alpha = 2.1$. As shown in Fig. 4, the normalized excess of the maximum amplitude $\Delta\alpha$ almost linearly increases in the range $1 \leq \alpha \leq 2$, whereas the increase in the phase shift $\Delta\theta$ slows down.

### 3.3 Interaction of internal waves with stable trapped cores

The large amplitude ISWs with $1.2 \lesssim \mathrm{Fr_{max}} \lesssim 1.3$ and $0.15 \lesssim \mathrm{Ri_{min}} < 1$ are characterized by stable long-lived cores. Fig. 6a shows the collision of waves with equal amplitude $\alpha = \alpha_L = \alpha_R = 3.3$ (case (A9; A9)) with the parameters $\mathrm{Fr_{max}}$=1.28 and $\mathrm{Ri_{min}}$=0.25. As seen in the figure, the volumes of dyed fluid in the trapped core collide together with the waves. The cores did not mix during the collision, which was also observed in a laboratory experiment (Honji et al., 1995). Then, the transmitted waves captured the cores and carried the dyed fluid in the opposite directions with little mass loss. Some mass exchange that occurred in the mode-2 experiment (Stamp and Jacka, 1995) was, perhaps, the result of a small offset pycnocline (Carr et al., 2015).



The interaction process is described in Fig. 7 in more detail. Here, the velocity and vorticity fields are shown together with an isopycnal distribution. At the beginning of collision (Fig. 7a), the trapped cores almost touch. They form a pair of vortices carrying trapped fluid upward. The next snapshot (b) corresponds to the time when the potential energy of the interacting waves reaches a maximum and the kinetic energy reaches a minimum. Unlike waves of class (i), at this moment in time, the kinetic

energy of the waves does not vanish because the flows in the trapped cores change sign when the colliding waves pass through each other. This process is also different from the process of the formation of waves with captured cores due to the collapse of the mixed region, which was initially in a state of rest. Then, the fluid in the cores is entrained by the transmitted waves with some mixing due to instability, resulting in the slight loss of mass from the trapped cores and a decrease in the phase velocity of 8 % (Figs. 7c and 6a). As shown in Fig. 4, for stable waves of class (ii), the runup excess $\Delta\alpha$ still almost linearly increases

in the range $2.3 \leq \alpha \leq 4.6$, whereas the increase in the phase shift $\Delta\theta$ is substantially slowed down when $\alpha > 1$, and then $\Delta\theta$ tends towards a constant value at $\alpha \geq 4$. The distributions of $\Delta\alpha$ and $\Delta\theta$ in Fig. 4 for stable waves were approximated based on the linear and exponential dependences, respectively, as

$$\Delta\alpha = 0.116\alpha, \quad \Delta\theta = 4.1[1 - \exp(-1.33\alpha)]. \tag{5}$$

The behaviour of mode-2 ISWs during reflection off a solid vertical wall is similar to that of the collision of two waves of

equal amplitude. A comparison with the simulated reflection of ISWs off a vertical wall (case D1) in a laboratory experiment (Stamp and Jacka, 1995) is given in Fig. 8. The parameters of the experiment were as follows: density difference $\Delta\rho/\rho_0$=0.05, pycnocline thickness $h = 0.0025$m and $\alpha = 2.2$. The incident wave belongs to the class (ii) ISWs (ii): $\mathrm{Fr_{max}}$=1.18, $\mathrm{Ri_{min}}$=1.05. The calculated density isopycnals in a vertical cross-section along the flume at $t = 16$s in Fig. 8a agree with the density isopycnals visualized in the experiment by water insoluble droplets of different densities in Fig. 8b. The interaction process

is similar to that shown in Fig. 6a, where after collision, some instability and mixing are observed in the rear of the trapped core. In simulated case D1, the corresponding values of $\mathrm{Fr_{max}}$ and $\mathrm{Ri_{min}}$ after reflection are 1.1 and 1.21, respectively. The simulated and observed trajectories of the wave crests, as shown in Fig. 8c, are similar. The corresponding simulated runup excess $\Delta\alpha$=0.28 and phase shift $\Delta\theta$= 3.8. These values agree with the other values of $\Delta\alpha$ and $\Delta\theta$ in Fig. 4. The experimentally observed (Stamp and Jacka, 1995) phase shift values are also given in Fig. 4b. They demonstrate large scatter due to difficulties

encountered in the experiment, as indicated by (Stamp and Jacka, 1995).

### 3.4   Interaction of internal waves with unstable trapped cores

The large amplitude ISWs with $\mathrm{Fr_{max}} \approx 1.3$ and $\mathrm{Ri_{min}} \lesssim 0.1$ belong to class (iii), which is characterized by a local wave-induced shear instability resulting in the appearance of the Kelvin-Helmholtz (KH) billows (Maderich et al., 2015); globally, however, this wave/self-generated shear system can be stable, as noted by (Almgren et al., 2012). The waves carry out trapped

fluid, but the cores gradually lose trapped fluid to the wake through KH billows shifting to the wave rear. Fig. 6b shows the collision of waves with equal amplitude $\alpha = \alpha_L = \alpha_R = 6.4$ (case (A13; A13), with the parameters $\mathrm{Fr_{max}}$=1.31 and $\mathrm{Ri_{min}}$ =0.06) for a 2D setting. Unlike that shown in Fig. 6a (case (A9; A9)), the collision of trapped cores was accompanied by billows, resulting in mixing. The divergent waves remained locally unstable, again forming KH billows in the wave aft. The



amplitude of diverging waves gradually decreased due to the loss of mass of the trapped cores. The mixing process caused by the destruction of the KH billows is essentially three-dimensional; therefore, 3D structures should be important for the transport of trapped cores (Deepwell and Stastna, 2016). The wave collision in a 3D setting for the same case (A13; A13) is shown in Fig. 9. The interacting ISWs are visualized by a density isosurface with $\rho$=1005 $kg\ m^{-3}$ at different times. In the side plans,

the distributions of the density are shown. Initially, the development of instability was two-dimensional (Fig. 9a), resulting in the development of KH billows similar to that in the 2D simulation in Fig. 6b. However, overturning the KH billows resulted in the appearance of spanwise structures. These processes were enhanced by the interaction of waves and their trapped cores, causing spanwise instability and turbulence (Fig. 9b,c). The diverging waves after the collision remained shear-unstable (Fig. 9d). The comparison of side plans for the 2D setting and for the cross-section averaged distribution of density for the 3D setting

are given in Figs. 9e and 9f, respectively. As seen in the figure, 3D instability results in greater mixing comparative with the 2D case. Therefore, KH billow visible in 2D setting (Fig. 9e) disappeared in cross averaged 3D distribution in Fig. 9f. However, differences between 2D and 3D estimations of $\Delta\alpha$ and $\Delta\theta$ in Fig. 3 did not exceed 2.5% and 1.2%, respectively.

### 3.5   Interaction of internal waves with trapped cores and different amplitudes

The collision process is modified for waves of different amplitudes by the exchange of trapped fluid between colliding waves

due to the conservation of momentum (Stamp and Jacka, 1995). This process is shown in Fig. 10 for two cases. In the first case, two stable strongly nonlinear waves with trapped cores collide ($\alpha_L = 3.3$ with $Fr_{max}$=1.28, $Ri_{min}$=0.25; $\alpha_R$ =2.15 with $Fr_{max}$=1.16, $Ri_{min}$ =0.4). As seen in Fig. 10a, the part of the trapped core fluid from the wave of larger amplitude (blue colour) is merged with the trapped fluid from the smaller wave (red colour) without noticeable mixing. The circulation inside the core of the larger transmitted wave results in the stirring of the fluid in such a way that the smaller core fluid eventually ends up

inside the fluid from the larger wave.

The collision of an ISW of small amplitude (class (i)) with a stable wave of large amplitude (class (ii)) was considered for case (A11; A1) to study the possibility of triggering instability in the wave of large amplitude via a small disturbance, similar to the waves of mode-1 in a two-layer fluid (Almgren et al., 2012). The simulation results are shown in Fig. 10b. As seen in the figure, the small amplitude ISW ($\alpha = 0.5$; $Fr_{max}$=0.33, $Ri_{min}$ =81) triggered instability in the ISW with an amplitude that was

ten times larger than that of the small wave. Notice that the large amplitude wave has parameters ($\alpha = 4.6$; $Fr_{max}$=1.3, $Ri_{min}$ =0.15) that are close to critical for the development of instability. The amplitude of the small wave essentially decreased during the interaction process due to the interaction and due to the viscous attenuation at the low Reynolds number of the wave ($Re_m$ =45.1). Unlike the head-on collision of large amplitude mode-1 and weakly nonlinear mode-2 ISWs (Stastna et al., 2015), the transmitted wave of small amplitude did not degenerate. Spatiotemporal diagrams for the paths of two ISWs of different

amplitudes colliding head-on are shown in Fig. 11 for cases (A9; A7) and (A1; A11). As seen in Fig. 11a, the trajectories of waves of larger amplitude propagating from left to the right were less subject to changes due to the collision, whereas the phase shift and the decrease of phase velocity for the smaller waves propagating from right to the left were essentially greater.



### 3.6 Estimation of the energy loss due to collision

The relative loss of energy due to the collision of ISWs can be calculated as the normalized difference in energy of incoming waves and transmitted waves

$$\Delta E = \frac{PSE_L^{(in)} + PSE_R^{(in)} - PSE_L^{(tr)} - PSE_R^{(tr)}}{PSE_L^{(in)} + PSE_R^{(in)}}, \tag{6}$$

where $PSE_L^{(in)}$ and $PSE_R^{(in)}$ are the pseudo-energies of the incoming waves at the cross-sections $x_L$ and $x_R$, respectively, and $PSE_L^{(tr)}$ and $PSE_R^{(tr)}$ are the pseudo-energies of the transmitted waves at the cross-sections $x_L$ and $x_R$, respectively. The pseudo-energy is the sum of the kinetic and available potential energies (Shepherd, 1993) of incident and transmitted waves. The method for estimation of the available potential energy and energy fluxes was given in (Scotti et al., 2006; Lamb, 2007). A detailed description of the procedure of the pseudo-energy calculation was presented by (Maderich et al., 2010). We define

the energy loss due to the wave collision ($\Delta E_{loss}$) as the difference between the total loss of energy $E_{tot}$ (6) and the loss of energy by single waves due to the viscous decay or instability $\Delta E_{dis}$

$$\Delta E_{loss} = \Delta E_{tot} - \Delta E_{dis}. \tag{7}$$

As seen in Fig. 12, the dependence of the relative loss of energy on the dimensionless wave amplitude for symmetric collisions ($\alpha = \alpha_L = \alpha_R$) is not monotonous. It can be divided into three different ranges. In range I ($0 \leq \alpha \leq 1$), the energy loss due

to the interaction increases as the wave amplitude increases. This range coincides with the range of weakly nonlinear waves without trapped cores. In range II ($1 < \alpha \leq 1.75$), the relative energy losses reach a maximum. The range coincides with the range in which colliding waves lose trapped cores in the process of interaction. This fact can explain the relative maximum of energy loss as the loss of potential energy of the cores. In range III ($1.75 \leq \alpha$), the behaviour of the loss of energy is also non-monotonous and non-similar. At first, in the zone of stable large amplitude collisions, the loss of energy decreases, but

as the amplitudes of collided waves increase, the interaction is accompanied by the development of instability; therefore, the loss of energy increases. Finally, for unstable waves, the energy losses due to the interaction increase monotonically with increasing amplitude. The Euler equations for stratified fluid do not contain non-dimensional parameters if the thickness of the stratified layer $h$, the phase velocity of the long linear waves $c_0$ and the characteristic time $\tau_0$ are used for the equation scaling. Therefore, it should be expected that the dimensionless relations in Figs. 4 and 12 will be similar for different $h$.

However, they demonstrate a lack of complete similarity due to the potential influence of other parameters. It was suggested by (Maderich et al., 2015) that this is a result of the incomplete similarity on the wave Reynolds number $\mathrm{Re_m}$, representing the effect of viscosity. As shown in Table 1, the parameter $\mathrm{Re_m}$ varies in Series A-C several times for waves of the same dimensionless amplitude $\alpha$. Another factor of the incomplete similarity can be the diffusivity effect (Deepwell and Stastna, 2016), which is described by the Schmidt number $\mathrm{Sc} = \nu/\chi$. However, in our study, the Schmidt number was constant.



## 4 Conclusions

The dynamics and energetics of a head-on collision of internal solitary waves (ISWs) with trapped cores propagating in thin pycnocline were studied numerically within the framework of the Navier-Stokes equations for a stratified fluid. The peculiarity of this collision is that it involves the trapped masses of a fluid. The interaction of ISWs differs for three classes of waves: (i) weakly nonlinear waves without trapped cores, (ii) stable strongly nonlinear waves with trapped cores, and (iii) shear unstable strongly nonlinear waves with trapped cores. The simulations showed that the wave phase shift grew as the amplitudes of the interacting waves increased for interacting waves of classes (i) and (ii) and remained almost constant for those of class (iii). The excess of the maximum runup amplitude over the sum of the amplitudes of colliding waves almost linearly increased as the amplitudes of the interacting waves belonging to classes (i) and (ii) increased. However, it decreased somewhat for those of the unstable class (iii). The dependence is similar to the interaction of the mode-1 waves (Terletska et al., 2017), with the difference being that the phase shift continues to grow for the collision of waves of mode-1. The waves of class (ii) with a normalized thickness of the pycnocline amplitude $\alpha$ fully lost fluid trapped by the wave cores in the approximate range of $1 \leq \alpha \leq 1.75$. The interacting stable waves of higher amplitude captured cores and carried trapped fluid in the opposite directions with little mass loss. The collision of locally shear unstable waves of class (iii) was accompanied by the development of three-dimensional instability and turbulence. The dependence of energy loss on wave amplitude was not monotonous. Initially, the energy loss due to the interaction increased with increasing wave amplitude. Then, the energy losses reached a maximum due to the loss of potential energy of the cores upon collision and then started to decrease. With further amplitude growth, the collision was accompanied by the development of instability, and the loss of energy increased. The collision process was modified for waves of different amplitudes because of the exchange of trapped fluid between colliding waves due to the conservation of momentum. Merging of the trapped fluid due to the collision of stable waves belonging to class (ii) occurred through the stirring mechanism without noticeable mixing. Similar to waves of mode-1 in a two-layer fluid (Almgren et al., 2012; Terletska et al., 2017), the interaction of a wave of large amplitude with a wave of small amplitude can trigger local wave instability of the large amplitude wave if the parameters of this wave are close to critical for the development of instability.

*Acknowledgements.* This work was supported by funds from CKJORC and the major research project of KIOST (PE99501).





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



**Table 1.** Summary of parameters of interacting ISWs

| Wave | $h$ (cm) | $a$ (cm) | $\lambda$ (cm) | $\varepsilon$ | $\alpha$ | $Fr_{max}$ | $Ri_{min}$ | $Re_m$ |
|------|------|------|------|------|------|------|------|------|
| A1 | 0.5 | 0.25 | 3.15 | 92 | 0.5 | 0.33 | 81 | 45.1 |
| A2 | 0.5 | 0.4 | 2.35 | 92 | 0.81 | 0.71 | 52 | 86.59 |
| A3 | 0.5 | 0.575 | 1.9 | 92 | 1.15 | 0.82 | 14 | 132 |
| A4 | 0.5 | 0.675 | 2 | 92 | 1.35 | 0.98 | 11 | 166.05 |
| A5 | 0.5 | 0.8 | 2.2 | 92 | 1.6 | 1.11 | 1.1 | 223 |
| A6 | 0.5 | 0.94 | 2.4 | 92 | 1.88 | 1.12 | 0.8 | 277.4 |
| A7 | 0.5 | 1.075 | 2.65 | 92 | 2.15 | 1.16 | 0.4 | 338.4 |
| A8 | 0.5 | 1.3 | 3.15 | 92 | 2.6 | 1.25 | 0.35 | 443.4 |
| A9 | 0.5 | 1.7 | 3.65 | 92 | 3.3 | 1.28 | 0.25 | 683.0 |
| A10 | 0.5 | 1.9 | 4.25 | 92 | 3.38 | 1.29 | 0.19 | 785.2 |
| A11 | 0.5 | 2.3 | 4.75 | 92 | 4.6 | 1.3 | 0.15 | 1075 |
| A12 | 0.5 | 2.5 | 5.35 | 92 | 5 | 1.35 | 0.12 | 1242 |
| A13 | 0.5 | 3.2 | 6.35 | 92 | 6.4 | 1.31 | 0.06 | 1681 |
| B1 | 1 | 0.63 | 5.21 | 46 | 0.63 | 0.51 | 24 | 153.4 |
| B2 | 1 | 0.85 | 4.23 | 46 | 0.85 | 0.68 | 11.5 | 225.5 |
| B3 | 1 | 1.25 | 3.61 | 46 | 1.25 | 1.02 | 2.4 | 388.0 |
| B4 | 1 | 1.95 | 5.25 | 46 | 1.95 | 1.16 | 0.38 | 765.5 |
| B5 | 1 | 2.68 | 6.65 | 46 | 2.68 | 1.22 | 0.18 | 1225. |
| B6 | 1 | 2.86 | 7.1 | 46 | 2.86 | 1.22 | 0.13 | 1345. |
| B7 | 1 | 3.56 | 8.6 | 46 | 3.56 | 1.23 | 0.11 | 1839. |
| C1 | 2 | 0.42 | 14 | 23 | 0.21 | 0.19 | 52 | 161.4 |
| C2 | 2 | 0.76 | 10.4 | 23 | 0.38 | 0.30 | 25 | 291.2 |
| C3 | 2 | 1.2 | 7 | 23 | 0.6 | 0.50 | 3.1 | 511.68 |
| C4 | 2 | 1.7 | 6.1 | 23 | 0.85 | 0.69 | 1.1 | 669.1 |
| C5 | 2 | 2.02 | 6.21 | 23 | 1.01 | 0.84 | 0.45 | 881.6 |
| C6 | 2 | 2.6 | 6.25 | 23 | 1.3 | 0.9 | 0.23 | 1159. |
| C7 | 2 | 2.9 | 7.22 | 23 | 1.45 | 1.01 | 0.22 | 1483. |
| C8 | 2 | 3.3 | 7.9 | 23 | 1.65 | 1.08 | 0.18 | 1851. |
| C9 | 2 | 3.5 | 8.5 | 23 | 1.75 | 1.147 | 0.15 | 2030. |
| C10 | 2 | 4.1 | 9.2 | 23 | 2.05 | 1.17 | 0.13 | 2521. |
| C11 | 2 | 4.56 | 10.82 | 23 | 2.28 | 1.23 | 0.12 | 2812. |
| C12 | 2 | 4.846 | 12.44 | 23 | 2.42 | 1.24 | 0.09 | 3171. |
| C13 | 2 | 5.28 | 13.4 | 23 | 2.64 | 1.25 | 0.07 | 3478. |
| C14 | 2 | 5.94 | 15.31 | 23 | 2.97 | 1.29 | 0.05 | 3884. |
| D1 | 0.25 | 0.55 | 12.5 | 56 | 2.2 | 1.18 | 1.05 | 329 |





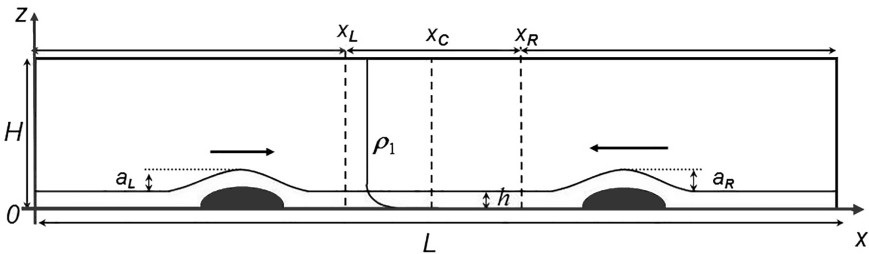

**Figure 1.** Configuration of the experiment exploring the interaction of ISWs with trapped cores.

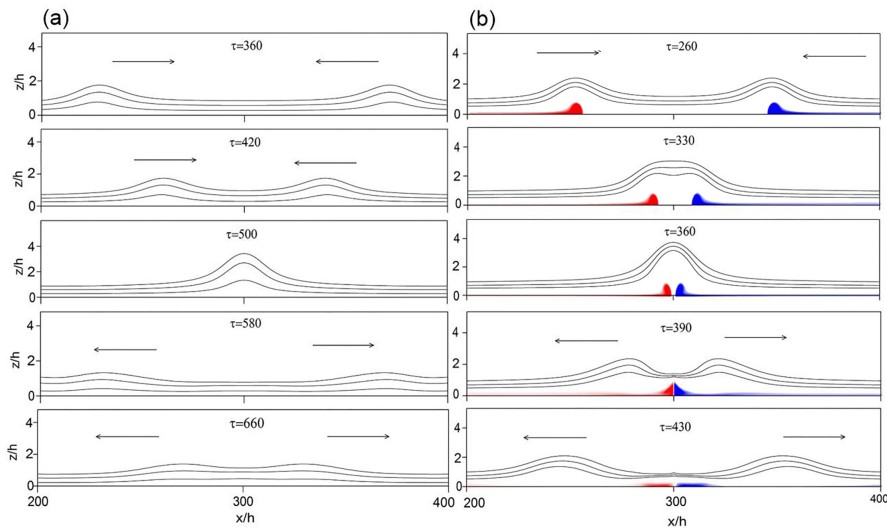

**Figure 2.** Snapshots of the density isopycnals during the collision of ISWs in a 2D setting. (a) Case (A2; A2). (b) Case (A5; A5). The trapped cores are visualized by dyed fluid.





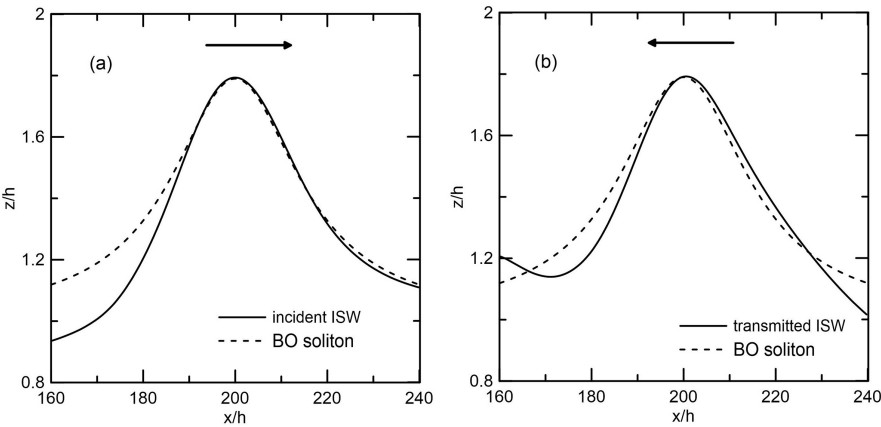

**Figure 3.** Wave profile of incident wave in section $x_L$ (a) and wave profile of transmitted wave in section $x_R$ (b) for $\alpha = 1.6$ (case A5; A5). These profiles are compared with the profile of the BO soliton.

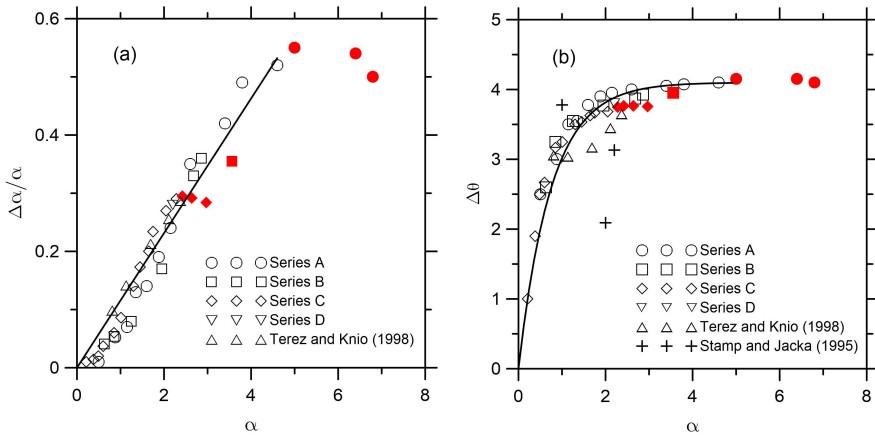

**Figure 4.** Relative runup excess $\Delta\alpha$ (a) and phase shift $\Delta\theta$ (b) of the interacting symmetric ISWs versus the normalized amplitude of the wave $\alpha$. The filled symbols correspond to the cases with KH instability. The fits, done by using a straight line in (a) and an exponential function in (b), are shown.

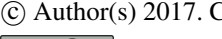



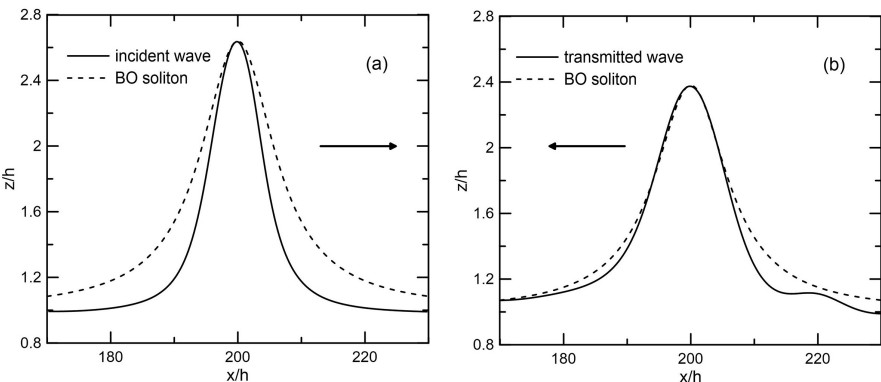

**Figure 5.** Wave profile of incident wave in section $x_L$ (a) and wave profile of transmitted wave in section $x_R$ (b) for $\alpha = 1.6$ (case A5; A5). These profiles are compared with the profile of the BO soliton.

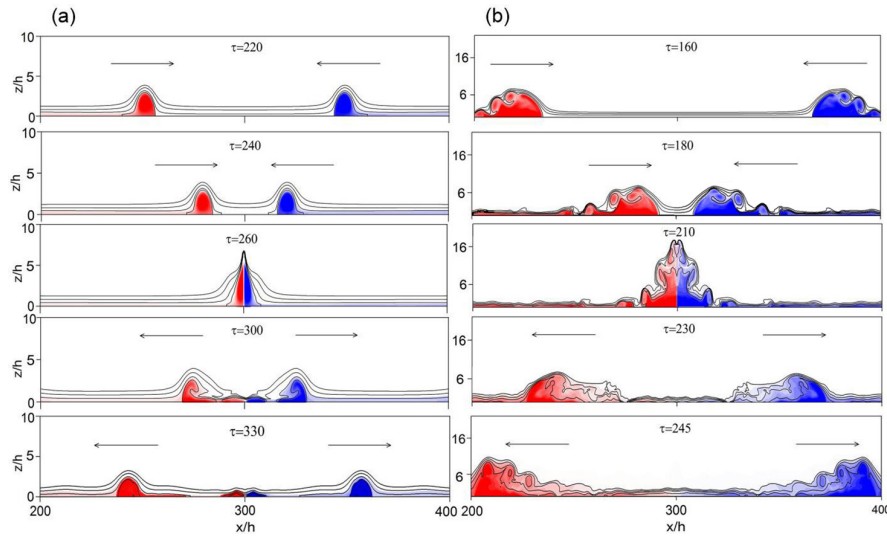

**Figure 6.** Snapshots of the density isopycnals during the collision of ISWs in a 2D setting. (a) Case (A9; A9). (b) Case (A13; A13). The trapped cores are visualized by dyed fluid.



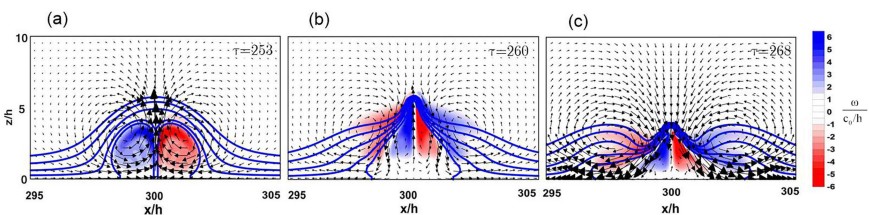

**Figure 7.** Details of the interaction of waves of equal amplitude $\alpha = 3.3$ at different times (case (A9; A9)) in a 2D setting. The velocity, vorticity $\omega$ and isopycnals are shown.

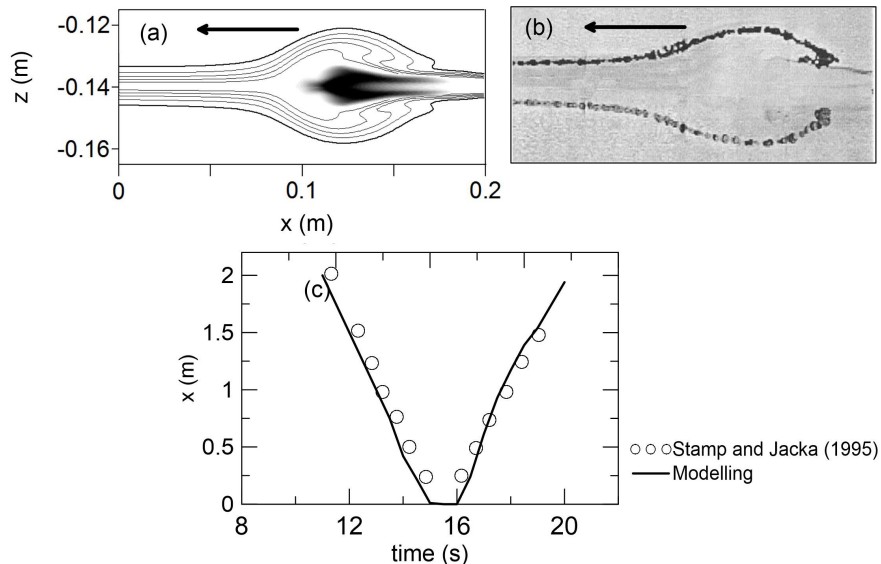

**Figure 8.** Comparison of the simulated reflection of ISWs off a vertical wall (case D1) in a 2D setting with a laboratory experiment (Stamp and Jacka, 1995). (a) Snapshot of the calculated density isopycnals, visualized by a black tracer trapped fluid at $t = 16$ s. (b) Density isopycnals in the experiment, visualized by water insoluble droplets of different densities. (c) Spatio-temporal diagrams of the path of an ISW during reflection off a wall.



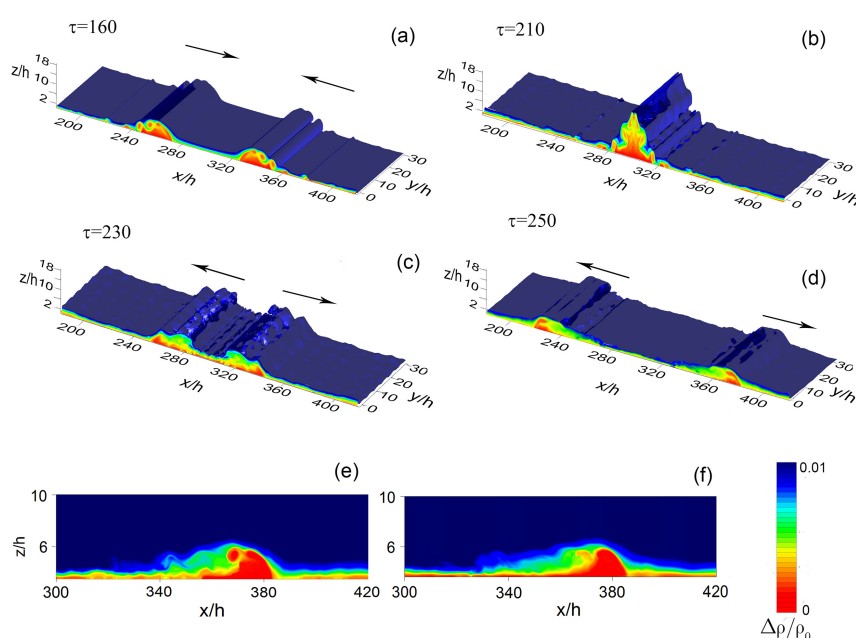

**Figure 9.** Three-dimensional evolution of ISWs for case (A13; A13) (a-d). The wave is visualized by a density isosurface with $\rho$=1005 kg m$^{-3}$. The side plan shows the distribution of density. The comparison of side plans for a 2D setting and for the cross-section averaged distribution of density for a 3D setting at $\tau = 250$ are given in (e) and (f), respectively.



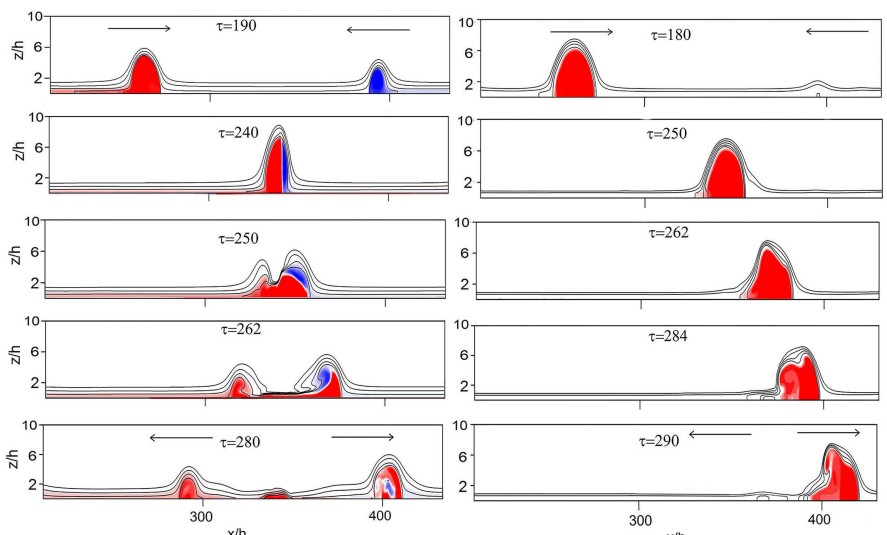

**Figure 10.** Snapshots of the density isopycnals during the collision of ISWs with different amplitudes for case (A9;A7) (a) and case (A11; A1) (b). The trapped cores are visualized by dye.

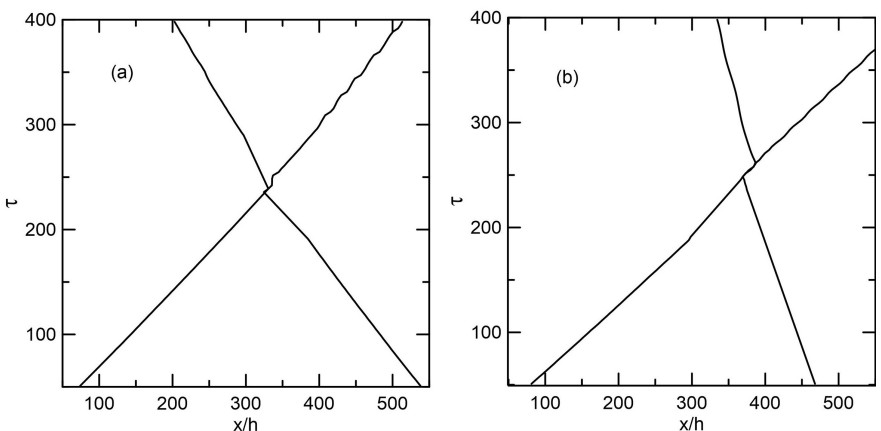

**Figure 11.** Spatio-temporal diagrams for paths of two ISWs of different amplitudes colliding head-on. (a) Case (A9; A7); (b) case (A11; A1).

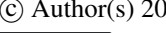



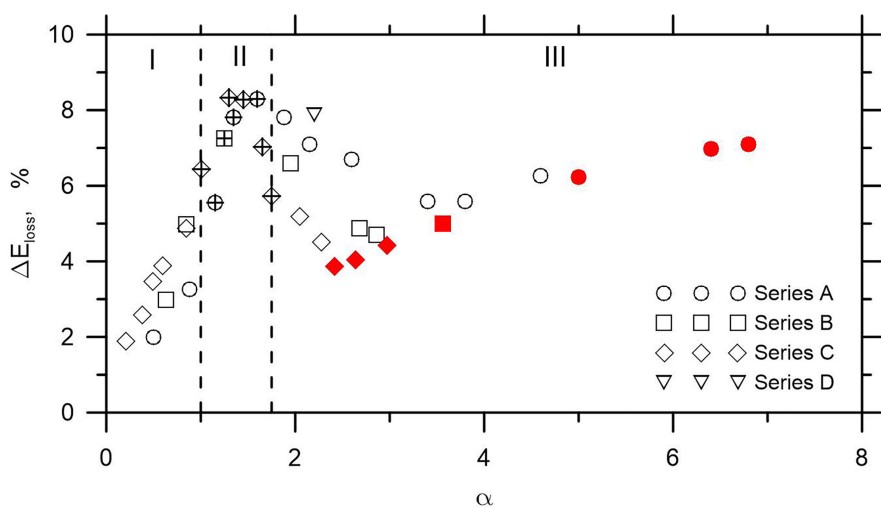

**Figure 12.** Plot of the energy loss versus the amplitude of the equal colliding waves. The filled symbols correspond to the cases with KH instability. The crossed symbols correspond to the cases where colliding waves lost trapped cores in the process of interaction.