# Peer review of "Head-on collision of internal waves with trapped cores"

_Nonlinear Processes in Geophysics, 2017_

## Referee Comment (RC1) · Anonymous Referee #1 · 12 Sep 2017

Please see attached .pdf document

Please also note the supplement to this comment:
https://www.nonlin-processes-geophys-discuss.net/npg-2017-31/npg-2017-31-RC1-supplement.pdf

---

## Referee Comment (RC2) · O. Fringer (Referee) · 20 Sep 2017

The comment was uploaded in the form of a supplement:
https://www.nonlin-processes-geophys-discuss.net/npg-2017-31/npg-2017-31-RC2-supplement.pdf

---

## Author Comment (AC1) · 30 Oct 2017

Dear Editor, enclosed are answers to comments by two reviewers . We have taken remarks by reviewers into account, and the paper has been revised accordingly. The main changes are: 1) 3D simulations were excluded for further separate publication; 2) The results of grid independence studies were discussed; 3) The discussion on effects of the Schmidt number was added; We look forward to hearing whether you can accept this article for publication.

Sincerely yours Vladimir Maderich October 30, 2017

Please also note the supplement to this comment:

[Figure]

https://www.nonlin-processes-geophys-discuss.net/npg-2017-31/npg-2017-31-AC1-supplement.zip

---

## Editor Comment (EC1) · M. Stastna (Editor) · 31 Oct 2017

The authors have responded to the reviewers in a reasonable way. I am thus prepared to receive what the authors feel is a nearly finalized manuscript. If the authors wish, I am willing to edit this version for technical English.

---

## Author Response (AR1)

**Reviewer#1.**

The authors are most grateful for your comments. We have followed your suggestions and revised the manuscript accordingly in many places. Please, find our responses below.

**GENERAL**

This paper uses primarily 2-D simulations to study the collision of internal solitary waves with trapped cores of different amplitudes. The motivation is observed collisions of Morning Glory clouds in Australia. Results focus on the phase shift, amplitude change and kinematic mechanisms underlying the actual collision. I find this paper to be an interesting read which, nevertheless, leaves several questions. Numerous questions exist about how the simulations sweep parameter space, how the initial trapped core waves are set up and the physical mechanisms behind the actual collision. In terms of the latter, I am greatly concerned about the adequacy of the 2-D and 3-D resolution of the simulations, particularly in light of the use of a Schmidt number of  $O(10^3)$  ?!? How well do these simulations resolve the finer features one expects, even in 2-D, due to the wind-up of the isopycnals by the K-H billows and how can we truly speak of turbulence and mixing at the resolutions used ? How much are the computed fields smeared at the finest-resolved scale by numerical diffusion ? Finally, there are a few points where the English needs polishing. One general grammatical comment: When describing the results, the authors often shift between past and present tense. Please keep the verb tenses consistent throughout the text. I list my specific comments below. If the authors address them I will gladly consider re-reading the paper to recommend it for publication.

Answer. See answers to specific comments.

**SPECIFIC**

**Abstract**

Line 12: Change "monotonous" to monotonic.

**Answer. Done.**

**Introduction**

Page 2, Line 2: The English feels awkward here. I would change to "... experiments and numerical solutions of both the DJL equation and the actual Navier-Stokes equations.

**Answer. Done.**

**Section 2**

**1.** Use of a Schmidt number of  $Sc = v/\approx 1,000$  is highly perplexing. Such a value of Sc should allow the formation of very fine scale patterns in the density field: 2-D runs can support very sharp gradients, either due to the straining of the pycnocline during collision or due to the roll-up of isopycnal lines by K-H instabilities, which are most likely below grid resolution. In 3-D, one would expect a Batchelor scale (presuming the K-H billows can attain some level of turbulence) which is equal to 10001/2 times smaller than the Kolmogorov scale. Are the simulations resolving this scale ?

The authors need to clarify the following points:

a. Have they conducted grid independence studies at least for their 2-D higher-amplitude ISW collision runs, where we expect the finest-scale patterns to form in the density field ?

**Answer.** We carried out doubling-grid tests to verify that chosen grid adequately described flow fields. The comparison for wave A13 is shown in Figs. A1 and A2 (see answers to Comments 1b-1c). The text was added accordingly.

p. 4 l. 25 "Most of the runs were performed in a two-dimensional setting with a grid resolution of  $3000 \times 400$  (length and height, respectively), whereas several runs for waves A9-A13 were also carried out with a grid resolution of  $6000 \times 800$  (length and height, respectively) to verify effect of grid resolution on the wave interaction and to make the fine structure clearer. Comparison of the baseline and doubled grid resolution showed the equivalence of the calculated fields, with the exception of wave A13 for which  $6000 \times 800$  resolution was used."

b. How many grid points span the actual pycnocline ? My back-of-the-envelope calculations show that the pycnocline is very coarsely resolved. Upon wave collision, it'll even be further strained and less resolved. Numerical diffusion of the low-order method underlying the authors' model can artificially smooth out things.

**Answer.** For the series A number of grid points span the pycnocline was 17 for grid 3000x400 and 35 for grid resolution 6000x800, for the series B the number of grid points span the pycnocline was 34 for grid 3000x400, whereas for the series C the number of grid points span the pycnocline was 68 for grid 3000x400.

c. In a 2-D run, how many grid points does one have across a K-H billow associated with instabilities along the wave ? One would need at least 30 grid points to guarantee that the resultant transverse instabilities are properly resolved in 3-D.

**Answer.** In our simulations about 45 grid points were placed across KH billow in the case (A13;A13) and Sc=1000 for grid resolution 3000x400 and more than 90 grid points covered KH billow for grid resolution 6000x800 as shown in Fig. A1-A2. For the rest of series of experiments this coverage was greater.

Fig. A1 Snapshot of the density field for case (A13;A13) at  $\tau = 175$  and Sc=1000 for grid resolution 3000x400 (a) and extended snapshot of KH billow with grid points (b).

Fig. A2 Snapshot of the density field for case (A13;A13) at  $\tau = 175$  and Sc=1000 for grid resolution 6000x800 (a) and extended snapshot of KH billow with grid points (b).

d. When 3-D runs are conducted, what is the local Reynolds number (based on local value of shear and B-V frequency along the wave-strained pycnocline) in the regions where K-H billows are observed, prior to K-H billow formation ? Is this Reynolds number high enough for actual turbulence to form within these billows or do they simply form, possibly pair and support some weak transverse instability ? How do we know that there are not scales smaller than the transverse instability that form ? Again, numerical diffusion can drive some very spurious results here.

Answer. We excluded results of 3D simulation from this paper.

e. MOST IMPORTANTLY: In 2-D, the authors should conduct a comparison of one simulation of high amplitude ISW collision at Sc = 1 and 1000, where I would hope/assume Sc = 1 is well-resolved by the authors' choice of grid. How do the results compare ? The Sc=1 case is presumably more relevant to the atmospheric Morning Glory case which motivates this study.

Answer. Text and figure were added to consider the impact of small diffusivity on the collision processes.

p. 7 1.8 "In the ocean and in the most of the laboratory experiments the Schmidt number is about 700-800. The used grid does not allow the whole range of inhomogeneities in salinity (density) to be resolved. Therefore, it is important to evaluate the effect of molecular diffusion of salinity on the dynamics of waves and to verify the possibility that diffusion can be neglected in the wave collision for large Sc. Two cases for large amplitude waves were considered (A9;A9) and (A13;A13). We performed runs for Sc=1; 10 and 1000. In the collision case (A9;A9) the behaviour of colliding waves are the same, whereas the difference between runs for Sc=1 and Sc=1000 was less than 1% of  $\Delta \alpha / \alpha$  and  $\Delta \theta$  values. The comparison of the density snapshots during collision in case (A13;A13) for different Schmidt numbers is shown in Fig. 9. Figure clearly depicts difference between structure of interacting waves for cases Sc=1 and Sc=10. The corresponding values of  $\Delta \alpha / \alpha$  and  $\Delta \theta$  differ by 5% and 0.6%, respectively. This was in agreement with the results by Deepwell and Stastna (2016), where it was shown essential effect of molecular diffusivity on the mass transport by mode-2 ISW in the range  $1 \le Sc < 20$ . At the same time, the results of calculations at Sc=10 and Sc=1000 in Fig.9b and 9c practically coincide, which indicates that molecular diffusion may not be taken into account when studying the global properties of colliding waves. This conclusion agrees with (Terez and Knio, 1998) as they estimate that the value of Sc=100 was "sufficiently high for density diffusion to be ignored during simulation period" and the results of the Deepwell and Stastna (2016) simulation, according to which the mass transfer is virtually independent of Sc already at Sc>20. However, diffusion can be important for small scale mixing processes in tiny density structures (see e.g. Galaktionov et al., 2001) forming in result of instability and turbulent cascade processes (Deepwell and Stastna, 2015) and persisting over time in a wake behind moving bulge of trapped fluid (Terez and Knio, 1998). These subgrid scale structures in our simulations were smashed by numerical diffusion which did not affect larger scale due to use of second order total variation diminishing (TVD) scheme for advective terms in transport equation. "